# Application of Data Modeling, Instrument Engineering and Nanomaterials in Selected Medid the Scientific Recinal Plant Tissue Culture

**DOI:** 10.3390/plants12071505

**Published:** 2023-03-30

**Authors:** Baoyu Ji, Liangshuang Xuan, Yunxiang Zhang, Wenrong Mu, Kee-Yoeup Paek, So-Young Park, Juan Wang, Wenyuan Gao

**Affiliations:** 1School of Pharmaceutical Science and Technology, Tianjin University, Tianjin 300072, China; 2Shool of Pharmacy, Henan University of Chinese Medicine, Zhengzhou 450046, China; 3Department of Horticultural Science, Chungbuk National University, Cheongju 28644, Republic of Korea

**Keywords:** bioreactors, intelligence model and optimization algorithm, multi-omics monitoring, nanomaterials, plant tissue culture, secondary metabolites

## Abstract

At present, most precious compounds are still obtained by plant cultivation such as ginsenosides, glycyrrhizic acid, and paclitaxel, which cannot be easily obtained by artificial synthesis. Plant tissue culture technology is the most commonly used biotechnology tool, which can be used for a variety of studies such as the production of natural compounds, functional gene research, plant micropropagation, plant breeding, and crop improvement. Tissue culture material is a basic and important part of this issue. The formation of different plant tissues and natural products is affected by growth conditions and endogenous substances. The accumulation of secondary metabolites are affected by plant tissue type, culture method, and environmental stress. Multi-domain technologies are developing rapidly, and they have made outstanding contributions to the application of plant tissue culture. The modes of action have their own characteristics, covering the whole process of plant tissue from the induction, culture, and production of natural secondary metabolites. This paper reviews the induction mechanism of different plant tissues and the application of multi-domain technologies such as artificial intelligence, biosensors, bioreactors, multi-omics monitoring, and nanomaterials in plant tissue culture and the production of secondary metabolites. This will help to improve the tissue culture technology of medicinal plants and increase the availability and the yield of natural metabolites.

## 1. Introduction

The increment in the global demand for medicinal plant resources, a diversified use of plants, and the reduction in cultivated land have accelerated the lack of plant resources. Large-scale culture of plant cells and tissues is considered to be a suitable method to alleviate this situation. In the past, due to the differences in plant species and varieties, the immaturity of culture conditions, and the solidification of research thinking, the development of plant cell and tissue culture technology was relatively slow [1]. Although the growth of plant tissues can be easily observed at different stages of in vitro culture, the growth environment of different tissues is nonlinear and uncertain, and is also affected by many other factors [2]. The complex interaction of many factors makes the use of traditional statistical methods problematic for optimization, and requires a certain amount of processing. New technologies have benefitted from the development of many fields such as information technology, engineering, instruments and the methods of analysis, and material science. In recent years, with breakthroughs in the biological mechanism of plant tissue and the continuous development of new technologies in various disciplines, the combination of theory and new technology has made rapid progress in the development of plant cell and tissue culture. Figure 1 shows the rapid progress in the development of plant cell and tissue culture with the help of the combination of theory and new technology. The development of various new technologies for plant tissue culture is aimed at achieving an efficient and high-volume production of plant biomass and effective secondary metabolites.

Whether for solid or liquid culture of plant tissue, the prediction and optimization of culture conditions have always been the focus of research. Data-driven modeling technology such as artificial intelligence (AI) [3], optimistic algorithm (OA) [4], and gene expression programming (GEP) can effectively be used for different purposes in plant tissue culture [5] such as the modeling, prediction, and optimization of plant genotypes, media, sterilization conditions, and plant growth regulators. Although we have these methods to optimize the culture condition, at present, the large-scale production of plant tissues can only be achieved by bioreactors. Bioreactors are engineering systems that support aerobic or anaerobic biochemical processes [6,7]. Various types of bioreactors have been designed and modified to improve the culture efficiency of different plant tissues. In addition, nuclear magnetic resonance-based metabolomics can be used as a tool to study the dynamics of plant cell metabolism and nutrition management [8]. Furthermore, nanomaterials also exhibit many special physical and chemical properties in plant tissue culture due to their unique structure and size [9]. Examples of the unique properties of nanoparticles include a very large specific surface area, high surface energy, and quantum confinement. These unusual properties may cause their environmental fates and behaviors to be very different to those of bulk congeners [10]. More and more nanomaterials are used in plant cell and tissue culture as elicitors [11,12]. Elicitors are often used in plant tissue culture to stimulate the defense system and the accumulation of secondary metabolites. In the past, biological elicitors and non-biological elicitors were more commonly used such as salicylic acid, methyl salicylate, bezoic acid, chitosan, bacterial, fungal, and algal [13,14]. In recent years, various types of nanomaterials have been used as new elicitors and have shown exciting effects in plant tissue culture [9]. Layered double hydroxide (LDH) [15] and chitosan [16] can be used as a transport carrier for various substances such as the transport of plant growth regulators, nutrients, etc., to promote the growth of plant tissue biomass and the contents of secondary metabolites. Engineered mesoporous silica nanoparticles (MSNPs) can be easily modified by specific functional groups, transported through the cell wall and cell membrane, and then enter the plant cells for the targeted binding of bioactive substances such as flavonoids [17]. After the combination, the MSNP-bioactive substance complex is carried out from the cell without causing significant harm to the hosts, achieving the sustainable harvesting of natural products. This paper summarizes the mechanism of medicinal plant tissue culture in recent years, and discusses the application of new technologies in plant cell and tissue culture.

## 2. Study on the Mechanism of Medicinal Plant Tissue Culture

### 2.1. The Generating Mechanism of Stem Cell, Callus, and Adventitious Roots 

During the late period of plant embryonic development, stem cells are located in the apical niche of meristem, and stem cells will divide and differentiate to maintain the constitution of meristem, which can grow into different tissues and organs such as callus, adventitious roots (ARs), etc. [18,19]. Root apical meristem (RAM) determines the embryonic development of the underground part of plants. The brassinosteroid (BR) has been shown to regulate stem cell niche (SCN) in RAM. In the BR signal module in RAM, the loss of functional BR leads to the decrease in cell division rate and the frequency of distal stem cell renewal [20]. Rape steroids (BRAVO) produced by the expression of R2-R3MYB transcription factors at the quiescent center (QC) play an important role in counteracting the effects of BR on cell division [21]. QC-expressed gene Wuschel-related homeobox 5 (*WOX5*) is crucial in the identity and maintenance of QC and the mutant *WOX5* showed an aberrant differentiation pattern of stem cells [22].

Callus induction is a key step in plant suspension cell culture such as *Zea mays* L. [23], *Panax ginseng* [24], and *Abrus precatorius* Linn [25]. The high concentration of indole-3-acetic acid (IAA) promotes root formation while the high proportion of cytokinin is good for adventive shoot regeneration [26,27]. Although the cell mechanism of callus induction is complex, we know that the continuation of the cell division cycle is the key to callus induction [28]. Somatic embryogenesis (SE) can follow two different pathways called direct and indirect SE. Direct pathways occur when plant cells produce embryos without forming calluses. The indirect pathway requires an additional step in callus formation before embryonic development [29]. Abnormalities in SE are associated with the use of 2,4-D in most published protocols, an inflammatory auxin that disrupts the balance of endogenous auxin and the auxin polar transportation interfering with the embryo apical-basal polarity [30]. Mechanical trauma may be the main inducing factors for organ regeneration, which triggers callus formation by the dynamic hormone level and transcriptional changes [31]. Callus formation is derived from J0121-labeled cells (J0121-label is a marker of the pericycle cells of xylem in roots and near xylem in aerial tissues), and the pericycle cells are considered to be the main contributors to the formation of callus. Pericycle cells can form callus or lateral roots (LR), adventitious roots, and other organs under certain conditions. An important transcription factor involved in the differentiation of pericycle cells is Miniyo, which may lead to rapid cell differentiation or prevent cell differentiation after entering the nucleus of pericycle cells [32,33] (see part a in Figure 2 for a detailed pathway).

Adventitious root (AR) cultures show the characteristics of high root proliferation and biomass, and have the potential to synthesize specific bioactive compounds. Adventitious roots are a reliable source of natural chemicals due to their genetic and biosynthetic stability. ARs can be induced from various explants (such as leaves, roots, stems, petiole callus, etc.) in in vitro conditions. Factors affecting the morphogenesis of ARs include indole-3-acetic acid (IAA), nitrous oxide, and light [34]. Among them, IAA is the main growth-promoting hormone that triggers AR; the fine-tuned spatiotemporal interactions between hormones ultimately regulate IAA distribution and perception [35]. In many plant species, higher IAA concentration is required in the early stage of adventitious root development than in the later stage. Some downstream signaling pathways in IAA transduction are involved in the molecular mechanism of adventitious root formation [36]. SLR is an auxin signaling factor in the Aux/IAA protein family. The lack of a functional SLR in the slr-1 dominant mutant resulted in reduced auxin response, affecting LRs and callus development. An IAA signaling molecule, ALF4 may regulate the stability of IAA28 protein, and increased SLR levels in ALF4 mutants may lead to callus and lateral root formation [37]. LR and AR share key elements of genetic and hormonal regulatory networks, but are still influenced by different regulatory mechanisms. Light is an important environmental parameter affecting the development of AR and LR. Photoreceptors are involved in regulating the development of AR and LR [38]. Studies on Arabidopsis show that roots have photoreceptors for blue, red, and far red light [39] as well as a potential interaction between light and auxin in AR regulation [40]. The growth response of roots to light and the initiation of AR may be related to the change in the local endogenous auxin concentration [41] (see part b in Figure 2 for the detailed pathway).

#### 2.1.1. The Mechanism of Endogenous Plant Growth Regulators on Plant Tissue Growth and Plant Stress Resistance

Plant development can be manipulated by adding PGRs at specific stages of growth or maturation by adding plant hormones such as IAA, naphthaleneacetic acid (NAA), 6-furfurylaminopurine (KT), 6-benzylaminopurine (6-BA), ethylene, abscisic acid (ABA), brassinolide (BL), jasmonic acid (JA), salicylic acid (SA), etc. [42]. All of these can stimulate further developmental responses. In particular, IAA and cytokinin play a decisive role in cell growth and differentiation (especially meristem differentiation) as well as apical dominance [19]. IAA as well as ethylene and cytokinin plays an important role in the formation of lateral roots [43,44]. The same growth regulator causes different responses to different explants or species, indicating that plant hormone receptors may be specific. Each hormone can correspond to multiple receptors in plants, and the relationship between plant hormone receptors is very complex [45]. The interaction of different hormone synthesis is also complex, as IAA can promote the biosynthesis of gibberellin (GA) to promote fiber development, and GA may regulate fiber development downstream of IAA [46]. The function and crosstalk of common hormones are shown in Table 1.

#### 2.1.2. Effects of Plant Growth Regulators on the Synthesis of Secondary Metabolites in Medicinal Plants

Plant secondary metabolites are the main active ingredients of pharmaceuticals. Plant growth regulators have been used in recent years as elicitors to stimulate the production of secondary metabolites. Plant growth regulators stimulate the synthesis of different types of secondary metabolites such as terpenoids and phenols (coumarins, lignins, flavonoids, isoflavones), tannins, sulfur containing secondary metabolites (glucosinolates, phytoalexins), and nitrogen containing secondary metabolites (alkaloids, cyanogenic glucosides and non-protein amino acids) [62]. Some studies have shown that PGRs are not as effective as traditional elicitors in improving the plant secondary metabolism level [63]. In addition to increasing biomass accumulation, PGRs will also induce the quantitative modification of major volatile components [64].

At present, although the specific mechanism of PGRs has not been elucidated, some studies have confirmed some parts of the mechanism of plant hormone response. Studies have shown that secondary metabolic clusters such as terpenoids share a common gene evolutionary history with the major metabolic pathways of plant growth hormone synthesis [64], which may reflect that the types of secondary metabolic compounds could be related to plant hormones. Plants transduce the signal of the nitrogen state through a variety of signaling pathways. One of the pathways is to use cytokinins as messengers. Cytokinin-mediated signal transduction is related to the control of plant development, protein synthesis, and macronutrient acquisition, and can coordinate with nitrate components to change plant metabolism [65]. Calcium–hormone interaction regulates the expression of phenylalanine ammonia lyase (PAL), stilbene synthase (STS), dihydroflavonol reductase (DFR), and UDP-glucose, flavonoid 3-O-glucosyltransferase (UFGT), which will increase the content of classified components (anthocyanins) in grape suspension cells [66]. IAA level can be upregulated under the combined stress of drought and salt. The contents of flavonoids and phenolic compounds as well as the transcriptional level and activities of enzymes related to secondary metabolism are also increased [67]. In addition to regulating the growth and development of plants, melatonin can also enhance the adaptability of plants to biotic and abiotic stresses. Its strong antioxidant capacity can enhance the ability of antioxidant enzymes in plants, thereby increasing the content of secondary metabolites. Furthermore, melatonin can reduce the stress response of plants under low iron or high iron conditions and upregulate the photosynthetic rate and synthesis of phenols and flavonoids [68].

#### 2.1.3. Biosensors Are Potential Tools for Plant Hormone Mechanism Research

A biosensor is an integrated receptor–transducer device that can convert a biological response into an electrical signal, biosensors for a wide range of applications such as health care and disease diagnosis, environmental monitoring, water and food quality monitoring, and drug delivery [69]. The interaction network of plant hormones is very complex. The visualization of the distribution and content of plant hormones in plant tissues can help people understand the basic laws of plant growth, and can also improve the yield and quality of plants by regulating the synthesis, transportation, and distribution of plant hormones in plants. The engineering of biosensors provides a new tool for the monitoring of phytohormones. The visualization and quantification of phytohormone distribution can be realized by using biosensors and other related technologies including fluorescent sensors, transcriptional reporters, degradation sensors, and luciferase [70].

Fluorescence biosensors are particularly effective for monitoring the spatial and temporal distribution of small signal molecules, which can be used to study the distribution and content of plant hormones in the subcellular [71]. The distribution of hormones in living tissues can be observed by the expression of the reporter gene. However, the relationship between the concentration of signal molecules and reporter molecules is complex, and it is necessary to use network dynamics to link the abundance of signals with the reporter’s signal [71]. Genetically encoded biosensors are able to detect rapid changes in the concentration and distribution of plant hormones in living cells [70]. Transcriptional reporters, degron-based biosensors, and FRET-based direct biosensors have been developed to monitor auxin. Ole Herud-Sikimić et al. [72] reported a binding bag specifically designed for auxin. When auxin binds to the bag, the conformation of the auxin binding part will be changed to couple with the fluorescent protein, then the fluorescence resonance energy transfer signal will be generated so that the auxin level can be reflected. Other types of sensors have been described. This article is no longer detailed [71].

## 3. Application of Technology with Different Fields

### 3.1. Using Data-Driven Modeling Technology to Optimize Tissue Culture Scheme

In order to successfully and efficiently carry out plant tissue culture, we need to scientifically improve the culture conditions including light, temperature and humidity, appropriate medium composition, plant growth regulators, etc. However, the cultivation of plants of different species and different tissues of the same plant has great differences. It is very tedious and inefficient to screen the most suitable culture conditions through the experiments. For example, in the prediction of a medium formula, the mineral and hormone composition of the plant micropropagation medium is an important factor in the growth of explants, but the different plant species lead to different nutritional and hormone requirements. Blindly testing the types and proportions of various hormones and minerals is time-consuming, laborious, and costly work, and does not often succeed. Most of the optimization of plant tissue culture medium is improved on the basis of previous experiments. This method requires a lot of control experiments. Second, it is also necessary to effectively develop and optimize the new medium. Understanding the role of minerals and hormones and their interaction with other medium components and different plant tissues is critical to the successful in vitro culture of plant tissues [73,74]. However, the relationship between medium components is very complex, and there are even many unknown interactions. It is difficult to optimize the medium for different plant species and genotypes. Therefore, the development and application of a predictive modeling system for medium formulation and culture conditions can improve the efficiency of culture optimization [74]. Data-driven modeling is considered as an effective alternative to optimizing biological processes and nonlinear multivariate modeling [75]. Modeling systems are often combined with optimization algorithms to improve efficiency. Appropriate modeling techniques can not only predict the value of the software sensor, but also estimate the probability [75]. In recent years, modeling technology has developed rapidly. Common systems include the combination of an artificial intelligence (AI) model and optimization algorithm (OA) [76], and the combination of gene expression programming (GEP) and genetic algorithm (GA) [5].

#### 3.1.1. Application of Artificial Intelligence (AI) Model and Optimization Algorithm (OA)

The modeling method of artificial intelligence (AI) is more effective than other modeling techniques and is a potential modeling tool for plant tissue culture [77]. AI and OA are widely used in different technical and scientific fields, and have been applied to improve different stages of plant tissue culture in recent years. Artificial intelligence tools build models from experimental and observational data. Using artificial intelligence models to model different steps of plant tissue culture is a very suitable and reliable method [75]. AI relies on the knowledge of mathematics and statistical equations and has engineering thinking and judgment capabilities. In recent years, various applications of AI and OA in the prediction of plant tissue culture medium and the optimization of growth and development have been reported. The steps of data modeling are: preprocessing of basic data (including principle component analysis and the data), network selection (parameter setting), training selection (error reduction) testing and interpretation of results, and the assessment of the developed model. Artificial intelligence tools include a variety of artificial neural networks, support vector machine (unsupervised learning artificial intelligence model for classification, clustering and regression analysis), and random forest (algorithms for classification and regression). All of which have the characteristics of simple design and high efficiency. The use of OA for optimal selection can significantly reduce the costs and time. This method relies on genetic algorithms and is especially suitable for plant tissue culture. The whole process takes into account the different culture stages of different plant tissues (such as embryo [75], callus [78], bud [79], and root [80]). The modeling, prediction, and optimization of plant genotypes, media, sterilization conditions, different types, and concentrations of plant growth regulators also need to be considered [75]. As shown in Figure 3, data-driven models are effectively used for different purposes in plant tissue culture. AI and OA have been effectively applied to predict and optimize the length and number of micro-buds [79,81] or roots [80], plant cell culture or hairy root biomass [82], and culture environmental conditions (such as temperature and sterilization [83,84]) to achieve the maximum productivity and efficiency as well as the classification of micro-buds and somatic embryos. Future AI–OA methods could also be used in the development of genetic engineering and genome editing.

#### 3.1.2. Application of Gene Expression Programming (GEP) and Genetic Algorithm (GA)

Previous studies have focused on predicting the effects of various components of the medium and hormones on plant explants through traditional multi-layer perceptron neural network (MLPNN) and multiple linear regression (MLR) methods. Genetic programming (GP) is one of the most traditional and widely used evolutionary algorithms. GEP technology involves computer programs encoded by linear chromosomes of different sizes and shapes, which is an effective alternative to traditional GP. Jamshidi et al. [85] used GEP and M5’ model tree algorithms to predict the effects of medium components on the in vitro proliferation rate (PR), branch length (SL), branch necrosis (STN), and vitrification (Vitri). This method mainly includes modeling systems, multiple linear regression, radial basis function neural network, gene expression programming, optimization of GEP models, and genetic algorithm model optimization. In addition, they also compared the methods of combining GEP with the radial basis function neural network (RBFNN) and multiple linear regression (MLR), respectively, to predict the effects of minerals and certain hormones in pear rootstock medium on the proliferation index. The results showed that RBFNN and GEP showed a higher accuracy than MLR. At present, the application of GEP in plant tissue culture is not replacing GA, but GEP has shown more accurate results in its application, and its further development has great potential. For the application of artificial intelligence and gene expression programming in tissue culture, see Table 2.

At present, data-driven modeling technology has become a predictive tool for modeling complex biological research and plays an important role in plant tissue culture. Scientific prediction greatly reduces the workload of researchers and reduces the cost of in vitro culture. In the future, the development of modeling technology may make the in vitro culture of plant tissue become more automated and mechanized.

### 3.2. New Technologies to Help Bioreactor Engineering

#### 3.2.1. Design of New Bioreactor

Many secondary metabolites of plants still cannot be obtained by artificial synthesis. The cell and tissue culture of medicinal plants is still the main source of secondary metabolite production [18]. Bioreactors can support the large-scale production of plant cells and tissues and has the advantages of stability, high efficiency, high yield, and low cost [6]. Bioreactors can be used to culture microorganisms or plant cells and tissues under monitored and controlled environmental conditions (such as pH, temperature, oxygen tension and nutrient supplement). Especially for medicinal plants with slow growth, a low bioactive component content, and those easily disturbed by environmental factors such as *Panax ginseng* and *Panax notoginseng*, the large-scale production of secondary metabolites by a bioreactor will be very competitive to solve existing problems such as the high cost of planting cultivation. By constructing the best environment for the effective growth and production of bioactive substances, the culture results can be significantly affected, which can be used to produce secondary metabolites, recombinant proteins, microbial fermentation, etc. From an economic point of view, the final purpose of the design of the bioreactor is to decrease the cost of planting production. To achieve this goal, a consideration of the cost of equipment, substrate cost, time, and the final total output is required [7].

Bioreactors should be able to control environmental conditions and allow the aseptic operation of different bioreactors to have a great influence on the state of secondary metabolites in plant tissue or cell culture. Agnieszka Szopa [87] compared the accumulation of phenolic acids and flavonoids in different types of bioreactors, and found that the conical bioreactor (CNB) was the best reactor for phenolic acid accumulation, and the nutrient sprinkle bioreactor (NSB) was the best for flavonoid accumulation. It can be seen that different reactors have different influences on the product selection. The appropriate design of the bioreactor will help to reduce costs, increase the productivity, and adaption to different kinds of cultures.

#### 3.2.2. Types of Bioreactors

Typical plant cell and tissue culture bioreactors are made of glass or stainless steel [88]. The morphology, rheology, growth, and production behavior of the culture should be considered comprehensively when selecting the most suitable bioreactor type. According to the state of the growth environment, bioreactors can be divided into liquid phase bioreactors, gas phase bioreactors, and gas phase and liquid phase combined bioreactors. According to the structure of culture materials, they can be divided into suspension cell culture bioreactors and tissue culture bioreactors. Liquid phase reactors include mechanically driven reactors and gas-driven reactors. The stirred reactor will cause damage to the culture materials. For gas-driven reactors, the oxygen flux and shear force should be considered [89]. In recent years, reactors dedicated to plant tissue and cell culture (PTCC) have been continuously improved including airlift, tubular membrane, silicone-tubing aerated, slug bubble, disposable wave and orbitally shaken, spray, helical ribbon impeller, rotating wall vessel (RWV) bioreactors, and stirred tank reactors [6]. Stirred reactors, rotating drum reactors, airlift reactors, bubble columns, fluidized bed reactors, and packed bed reactors are the most commonly used industrial and commercial reactors [88]. The advantages and disadvantages of a variety of reactors have been summarized by our predecessors, so will not be elaborated here. Currently, one problem is the possibility of microbial and/or viral contamination in bioreactors. These can occur by accident at any step in the culture process, and the contamination can be bacteria, mycoplasma, viruses, parasites, and fungi. In general, aseptic techniques, fungicides, and or antibiotics can be used to prevent contamination from occurring. The current work is focused on detecting bacterial contamination in bioreactors [90].

Stirred tank reactors (STRs) are currently the most functional bioreactors in PTCC. Airlift and bubble column systems were basically designed to improve the biological oxygen demand of the culture. Helical and double helical ribbon impellers appeared to be efficient for the suspension culture of high-density plants. An important advance in plant cell and tissue culture is the use of disposable bioreactors. A disposable reactor replaces the stainless-steel material in the device with plastic products, which greatly reduces the preparation time and cost of the reactor. Another advantage of disposable bioreactors is that they do not require cleaning, which will reduce the probability of contamination. At present, due to limited experience in the use of disposable reactors, a lack in the physical strength of plastic materials, and other reasons, this type of reactor is not widely used [91]. A large number of condition optimization experiments can be carried out simultaneously by using laboratory glassware or designing small reactors. The laboratory optimization of reactor culture conditions include temperature, pH value, oxygen supply, solid–liquid ratio, medium composition, and elicitor.

#### 3.2.3. Dynamic Monitoring during Scale-Up Culture

A variety of detecting techniques for omics research such as high-performance liquid phase (HPLC), mass spectrum (MS), and nuclear magnetic resonance (NMR) can be used to determine the type and content of animal and plant metabolites. Some have been used to monitor robust and high-yield biological production processes. The conventional physical parameters that should be detected are pH, temperature, and dissolved oxygen. Sensors for bioreactor detection need to accurately measure the concentration of various nutrients and metabolites in the culture medium without affecting the normal production process. For nutrients and metabolites in reactors, spectroscopy is currently used for monitoring. Spectroscopy is particularly important for the biotechnology industry because it can be easily conducted and achieve online real-time measurement. The overview of spectroscopy is to study the interaction between matter and electromagnetic radiation. Analytical instruments include infrared spectroscopy, fluorescence spectroscopy, etc. Finally, statistical and mathematical techniques are used to analyze the chemical data of the monitored material. Each spectral method has its own advantages and disadvantages, which means that different spectral methods have different components and bioreactor matrices to analyze. Spectroscopy for bioreactor monitoring has been used in microbial fermentation, cell culture, etc. [92]. NMR is an important analytical technique in material science and medicine. NMR-based metabolomics is used to monitor the concentration of metabolites and can be used as a tool to study the dynamics of plant cell metabolism and nutrition management. The advantages of NMR include fast, reliable, and allow for online measurements of cell media, etc. [93]. Ninad Mehendale [94] proposed an NMR compatible platform for automatic real-time monitoring of biochemical reactions using a flow shuttle configuration for the real-time monitoring of biochemical reactions. Another advantage of the proposed low-cost platform is high spectral resolution. Nuclear magnetic resonance metabolomics can monitor metabolite reactions in different optimization processes.

### 3.3. Application of Nanomaterials in Plant Tissue Culture

Nanomaterials have a wide range of applications in biotechnology, and have played a great role in medical, agricultural, pharmaceutical, and other fields. They can be used in sensor development, agricultural chemical degradation, soil remediation, drug delivery, and so on. Various materials are used to manufacture NPs such as metal oxides, ceramics, silicates, magnetic materials, semiconductor quantum dots (QDs), lipids, and polymers. Dendritic macromolecules and emulsion encapsulated polymer nanomaterials have controlled and distressed release capabilities, and metal-based nanomaterials have a size dependence. Nanomaterial formulations increase system activity due to higher surface area, solubility, and smaller particle size [95]. In the field of plant tissue culture, nanomaterials are widely used as elicitor stimulation to increase the accumulation of secondary metabolites [96], or as a transport carrier for plant hormones [97] and other substances to specifically harvest effective substances in plant cells [17].

#### 3.3.1. Nano-Elicitor

##### The Mechanism of Action of Nano-Elicitor

An elicitor stimulates the accumulation of secondary metabolites by stimulating the defense system in plant metabolism. Biological elicitors and abiotic elicitors are commonly used in plant tissue culture. Nanomaterials have been widely used in medicine, energy, biotechnology, and other fields due to their unique properties. In recent years, studies have found that using nanomaterials as elicitors can greatly increase the accumulation of secondary metabolites. The properties of nanomaterials can be precisely controlled by controlling the shape, size, and chemical composition of the materials. Nanoparticles can increase the production of reactive oxygen species and hydroxyl radicals that distort cell membranes, leading to changes in penetrability, facilitating the entry of nanoparticles into plant cells, and stimulating the production of secondary metabolites [98]. The process of nanomaterials acting on plant cells is shown in Figure 4. Controlled release of active ingredients is due to the slow release characteristics of nanomaterials, the combination of ingredients and materials, and the control of environmental conditions [95].

##### Application of Nano-Elicitors

Common nanomaterials include multiwalled carbon nanotubes (MWCNTs), single walled carbon nanotubes (SWCNTs), graphene, and fullerene C70 [99]. However, not all nanomaterials can be used as plant tissue culture elicitors. In addition to the reported common metal nanomaterials (such as AgNPs, ZnNPs, CuNPs, TiNPs, etc.), there are other new nanomaterials used in plant tissue culture. Carbon nanotube nanomaterials and multi-walled carbon nanotubes can induce about twice the increase in the total phenol content (TPC) in thyme (seedlings); for flavonoids (TFC), this number would be 1.09-fold [100]. Studies have shown that the localization of graphene nanomaterial (GNS) inside the chloroplasts may be to activate photosynthetic pigments, thereby exhibiting a stimulating effect on fructose, sucrose, and starch. This will increase the pepper and eggplant yields, which will have no bad effect on the plants themselves. Nanomaterials can be combined with a variety of chemical functional groups to form composite nanomaterials, which indicates better influences on the content of secondary metabolites. The effects of silver nanoparticles, graphene, and their nanocomposite nanoparticles on stevia plants have been studied. Plants treated with nanocomposites were found to have more stevioside and rebaudioside [99]. At the same time, the effects of MgONPs, perlite, and composite nanomaterials made of MgONPs, perlite, and plant extracts on the volatile content of *Melissa officinalis* leaf were compared, and it was found that the composite elicitor could increase the content of the product by 0–17% [101].

Nano-elicitors play an important role in the production of many natural products. The discovery and preparation of nanomaterials are key technologies. Therefore, which nanomaterials can be used as potential elicitors for plant tissue culture? To answer this question, we conducted the following analyses. First, due to the application of new nanomaterials with the development of nanotechnology, more and more new nanomaterials have been created, so there is a need to combine biotechnology and engineering to achieve multidisciplinary cooperation and development. Second, although some composite nanomaterials have not been used in plant tissue culture, they have potential capabilities such as natural polysaccharide composite nanomaterials [102]. This material contains naturally extracted chemical components that are safer for plants, and this composite material generally has antibacterial and antioxidant functions that can reduce bacterial contamination in plant tissue culture and achieve a greener production model. Third, simple nanomaterials that bind to biological or abiotic elicitors [103] also have antibacterial ability, and it is also convenient to conduct parallel control experiments with common elicitors to obtain a better one. The reported effects of different types of nano-elicitors on plant tissues are shown in Table 3.

#### 3.3.2. Nanomaterials as Transport Carriers

Nano-transport technology can transport some chemicals into cells in a targeted and efficient manner, which can effectively reduce the environmental pollution caused by plant growth. Currently, inorganic nanoparticles have been studied for drug delivery applications [113]. Nano-fertilizers and nano-insecticides have increased the economic benefits of agricultural products by 20–30% [114,115]. Nanomaterials for drug delivery such as calcium phosphate, gold, carbon materials, silicon oxide, iron oxide, and lactate dehydrogenase have multifunctional properties suitable for cell delivery such as wide availability, rich surface function, good biocompatibility, potential target delivery ability, and controllable drug release ability [116]. Plant development can be manipulated by adding plant growth regulators at a specific stage of growth or development, and further developmental responses can be stimulated by adding growth regulators such as auxin, cytokinin, gibberellin, ethylene, or abscisic acid. Auxin and cytokinin are the most important of these growth regulators. However, because these substances are greatly affected by the plant growth environment and the species itself, their content in plants is very unstable. On the other hand, PGRs are easily degraded by light, temperature, and other environmental factors, resulting in the loss of their activity, thereby affecting the growth of plant organs [97]. To promote the growth of plant tissue culture materials, we need to find a more efficient application model to ensure an even distribution of plant growth regulators in plant tissues. At the same time, we also need to supply plant growth regulators according to the needs of the plant tissue to achieve the rapid accumulation of plant tissue biomass [117]. Nano-carriers have great prospects for the application of plant growth regulators such as layered double hydroxide (LDH) [15] and chitosan nanoparticles [16].

##### Layered Double Hydroxide Nanomaterials as Transport Carriers

LDH has certain advantages as transport carriers. The hydroxide component of LDH can effectively prevent external enzymes and oxygen from destroying the loaded drugs in the interlayer. The intercalated structure makes the drugs carried by LDH have a controlled release effect. LDH-drug mixed nanoparticles can achieve cell localization by adjusting the zeta potential. LDH material has low cytotoxicity [116], and its size should be kept below 150–200 nm for endocytosis [118]. LDH has been used as a gene and in drug delivery in recent years [119] such as in an antioxidant carrier, fertilizer carrier [119], plant nutriment [15], pesticide slow release carrier [120], herbicide carrier [121], and plant hormone transport carriers.

Ilya Shlar et al. [116] found that LDH could be used as the carrier of IAA and acted on the plants of *Vigna radiata* (L.) Wilczek. The molecules of IAA are effectively inserted into LDH through the co-precipitation effect. The insertion can protect IAA molecules from enzymatic degradation, and the intercalation can achieve a sustained release performance. The results showed that the IAA–LDH complex treatment increased the rooting efficiency by 4.6 times and improved the biological activity of IAA to promote adventitious root development. Hussein et al. [122] found that the use of zinc-aluminum layered double hydroxide nanocomposites could achieve the controlled release of plant growth regulator NAA in the film. Yanfang Liu et al. [123] used the ion exchange method to insert sodium naphthalene acetate (NAA, a plant growth regulator) into layered double hydroxide Mg/Al-LDH, so that NAA showed a significant release effect. Vander A. de Castro et al. [117] used an alginate polymer (a substance with good biodegradability) to encapsulate zinc-aluminum LDH and NAA to act on plant seeds. The results showed that the alginate film containing ZnAl–NAA–LDH enhanced the root area, fresh root material, and shoot length of the plants. Inas H. Hafez et al. [124] constructed a gibberellic acid (GA) nanohybrid system using inorganic magnesium aluminum layered double hydroxide (LDH) as raw material. The biodegradation process of intercalated GA is characterized by a long soil storage period and slow degradation rate. Shifeng Li et al. [125] prepared β-naphthoxyacetic acid (BNOA) layered double hydroxides (LDHs) by the co-precipitation method and discussed its release mechanism. They believed that with the cleavage of the LDH nanolayer structure, the nanohybrid of BNOA-LDHs has good controlled release characteristics, and pH is the key factor.

##### Chitosan Nanoparticles as a Transport Carrier of Substances in Plant Tissue Culture

Compared with synthetic polymers, natural polymers have good biocompatibility and biodegradability. The structure of some natural polymers is similar to that of biological macromolecules, and they are more easily recognized, utilized, and metabolized by organisms [126]. Chitosan is obtained by the acetylation of chitin after alkali treatment. The structure of the amino group and carboxyl group can make chitosan functionalized such as carboxymethylation, etherification, esterification, crosslinking copolymerization, and other modifications [127]. Research has found that chitosan has good biodegradability, bioadhesion, ecological safety, and high biocompatibility due to its high charge density, the interaction of amine and carboxyl groups, and the existence of hydrogen bonds [128]. With these properties, chitosan can be used in biotechnology, medicine, food, agriculture, environment, and other fields [129]. Chitosan nanocarrier systems have also been developed for use in plant growth regulators as chitosan molecules can slowly release plant growth regulators. There are many benefits of controlled release such as protecting plant hormones from the environment and protecting plant cells from the risk of explosive release [16].

Chitosan nanomaterials have the characteristics of promoting plant growth and increasing the content of secondary metabolites. Farhad Mirheidari et al. [130] applied IAA, GA3, and chitosan nanofibers (alone or in combination) to *Roselle* plants. The results showed that IAA + GA3 + CNF (800 + 800 + 100 mg/L) stimulated the growth parameters of rose plants and promoted the content of healthy phytochemicals (ascorbic acid, β-carotene, anthocyanin) and the improvement in plant antioxidant capacity. This indicates that the combined application of plant growth regulators and chitosan nanofibers (CNFs) has an important impact on various growth parameters and metabolite status, and reflects the positive effect of chitosan nanofibers on plants. In plant tissue culture, chitosan is mostly used as a transport carrier for plant hormones. Salicylic acid-chitosan nanoparticles ensure their continuous availability in plants by slowly releasing SA. The seedling index was 1.6 times higher than that of the control group, and the chlorophyll (a,b) content (1.46 times), ear length, grain number per ear, and grain weight per pot were also increased [131]. Plant hormones such as SA can fight toxic symptoms caused by heavy metal stress such as chlorophyll synthesis and biomass loss [132]. Using SA-loaded chitosan nanoparticles to act on plants can promote the absorption of SA by plants. Studies have shown that salicylic acid nanoparticles (SANP) improve plant growth and phytoremediation efficiency under arsenic stress, and enhance the ability of plants to resist arsenic stress [133]. As an essential element for plant growth, Cu may also cause heavy metal toxicity to plants. Any organic compound that can affect plant development can be seen as a plant growth regulator. It has been proven that the complex of 1-hydroxy-1-methoxycarbonyl-copper (Cu(II)) can be used as a new plant growth regulator. This regulator can realize the controlled release by delivering the Cu(II) complex to plants using chitosan-coated calcium alginate microcapsules [134]. Some chitosan nanoparticles can also be used as carriers of bacteria. J. J. Perez et al. [135] developed a novel, green, low-cost chitosan-starch hydrogel as a delivery vehicle for plant growth-promoting bacteria—*Azospirillum*. The results showed that the release of bacteria in saline was gradual and could be used as a bio-nano fertilizer. Different chitosan nanomaterials as plant tissue culture material transport carriers are shown in Table 4.

#### 3.3.3. Nano-Assisted Harvesting Technology

Plant secondary metabolites are synthesized and accumulated in plant cells, but traditional recovery methods generally dry plant tissues, which is destructive to the tissue. In addition to destroying expensive transgenic plant cell cultures, the activity of unstable biomolecules may be lost during the solvent extraction process, resulting in reduced yields [141]. Nano-harvesting refers to the use of nanoparticles to combine and carry active molecules away from plant cells. This method can not only obtain secondary metabolites, but also protect the integrity of the original cells and is a sustainable method for harvesting metabolites from plant tissues. Engineered mesoporous silica nanoparticles (MSNPs) have the characteristics of high surface area, unique size, shape, pore structure, and surface functionalization. Such properties allow MSNPs to be easily modified by specific functional groups to target binding bioactive materials through cell barriers. Mesoporous silica nanoparticles designed with an amine function can bind to active substances of flavonoids and carry them out of the cell [17]. Silica nanoparticles were found to obtain flavonoids from plant cultures without significant harm to host plants [142]. After the harvest of nanoparticles, the roots were found to be re-synthesized by regulation [142].

The cellular uptake and excretion mechanisms of MSNPs are important in designing novel biomolecule separation and delivery applications. Positively charged MSNP particles have been shown to facilitate penetration into the membranous and cell walls. The amine-functionalized MSNPs spontaneously entered and exited the plant cells through dynamic exchange for 20 ± 5 min. Ti-functionalized weakly charged MSNPs were absorbed and excreted through a thermal activation mechanism, whereas amine-modified positively charged particles were absorbed and excreted mainly through direct cell penetration. Particle size and surface properties (charge) are the two most critical factors for cellular uptake, but the extent of uptake depends on the type of plant. Excretion mechanisms depend on the cell type, nanoparticle size, shape, and surface modification, and the surface charge may be a decisive factor in controlling transport and excretion [17].

The nano-capture of secondary metabolites is a new biotechnology for the separation and transfer of active biological molecules in living tissues. At present, nano-harvesting technology is only applied to the capture of flavonoids, but through the modification of nanomaterials, other active ingredients are also expected to be extracted.

## 4. Conclusions

The cultivation technology of plant cells, tissues, and organs is a modern biological means to solve the shortage of medicinal resources and the shortage of wild plant resources. The ultimate goal of plant tissue culture is to achieve large-scale biomass production and the accumulation of precious secondary metabolites. The mechanism of plant tissue growth and development is constantly being updated, and the technology in various fields is gradually being integrated into tissue culture technology. For example, data-driven modeling technology takes artificial intelligence modeling and optimization algorithm as the means of prediction, accurately and efficiently analyzes the cultivation conditions, develops new cultivation methods, and systematically and scientifically optimizes the prediction. Data-driven modeling is one of the most applicable methods and plays an important role as a predictive tool for modeling complex biological research. From another aspect, the amplification of the bioreactor culture and improvement in the dynamic monitoring system are conducive to the realization of the industrial automatic production of culture. Designing different bioreactor types will not only help to improve and maintain high productivity, but also reduce the process costs. Suitable types of reactors should be selected for different cultures. Nanotechnology has been widely used in plant tissue culture. However, nanotechnology is not limited to the content introduced in this article. Its mode of action is more extensive and can be applied in various processes of plant tissue culture. The role of nanomaterials largely depends on their chemical and mineral composition, size, and sometimes the shape and concentration of application. Technology in various fields with continuous update, and more innovative ways will be applied to the plant tissue culture process. In summary, the application of new technologies in intelligent algorithms, instrument engineering, nanomaterials, and other fields to adapt to the standardization of commercial applications has had a positive impact on plant tissue culture and the industrial production of secondary metabolites.

## Figures and Tables

**Figure 1 plants-12-01505-f001:**
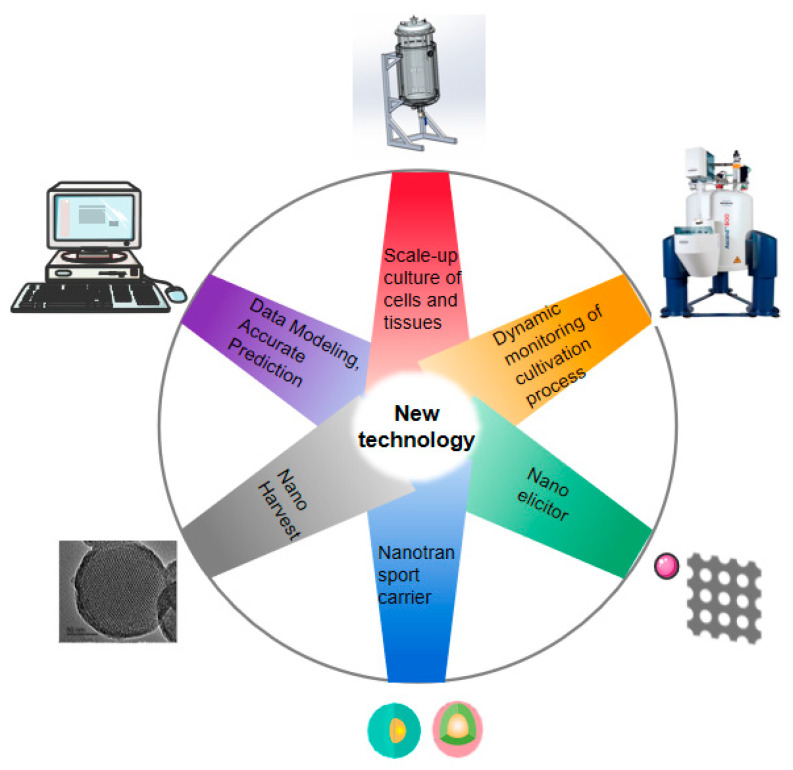
New technologies for plant tissue culture.

**Figure 2 plants-12-01505-f002:**
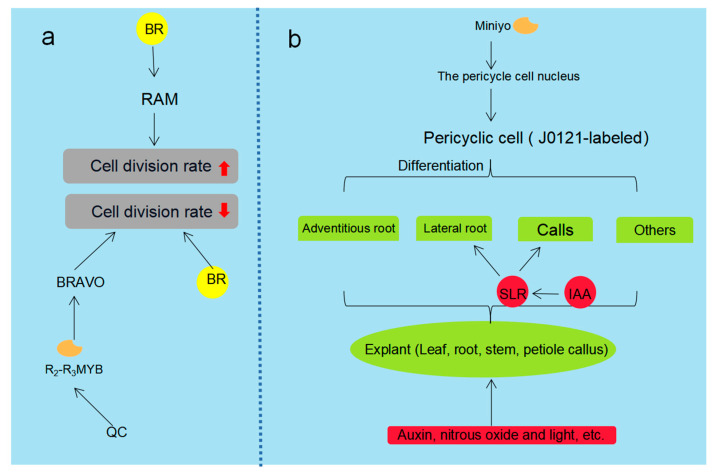
Effects of plant growth regulators. Note: (**a**) The effect of BR and transcription factor s on cell division rate. (**b**) Different plant tissues are differentiated from pericycle cells or induced by plant hormones. Yellow represents exogenous hormones, red represents exogenous hormones.

**Figure 3 plants-12-01505-f003:**
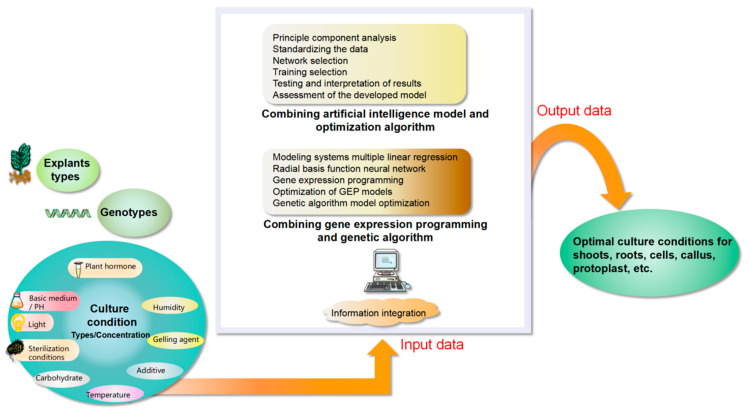
Overall roadmap for data-driven modeling technology.

**Figure 4 plants-12-01505-f004:**
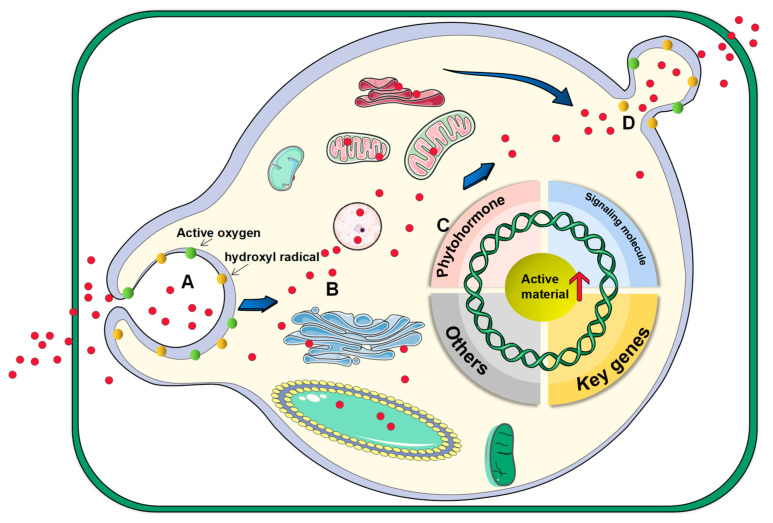
The process by which nano-elicitors act on plant cells. (**A**): The nano-elicitor increases reactive oxygen species and hydroxyl radicals, which can enhance the permeability of cell membranes. (**B**): The elicitor enters the cytoplasm and various organelles. (**C**): The elicitor stimulates intracellular hormones, related signaling molecules, key genes, and other components related to the synthesis of active substances, ultimately leading to an increase in the production of active substances. (**D**): Nano-elicitors are excluded from the cell by exocytosis.

**Table 1 plants-12-01505-t001:** Function and crosstalk of common hormones.

Plant-Growth Regulator	Classification	Function	Hormone Crosstalk
Auxin	IBA	Root contact induction [47]	The hypocotyl and is enhanced by application of (IBA) combined with kinetin (Kin) [48]
	IAA	IAA affects plant growth and development including growth response, vascular development, leaf and flower initiation, root growth, and lateral root formation [49]	GA enhances auxin levels in stems by stimulating polar transport of IAA [50]
Cytokinin	6-BA	Involved in cell division [51]	2,4-dichlorophenoxyacetic acid (2,4-D), indoleacetic acid (IAA), and 6-BA promote the production of somatic cells [52]
Gibberellins	GA	Promote germination, growth and flowering, promote leaf expansion, but inhibit root growth [53]	GA and cytokinins antagonize many developmental processes including shoot and root elongation, cell differentiation, shoot regeneration in culture and meristem activity [49]
	SA	Depending on its concentration and plant growth conditions and developmental stages [54]	SA regulates IAA biosynthesis and transport. Low concentration of SA (50 μM) promotes adventitious roots and changes the structure of root apical meristem [55]
	ABA	Regulate seed dormancy and germination [56]	ABA, SA, and auxin can increase plant resistance to pathogens [57]
	Ethylene	Branch elongation and leaf abscission [58]	The signaling mechanisms of gibberellin, ethylene, and brassinolide may not evolve until mosses and vascular plants have evolved [59,60]
	Brassinolide	Can promote the growth of plant seedlings	Working with other PGRs alone affects plant growth and development and abiotic and biotic stress responses such as ABA, ethylene, SA, JA [61]

**Table 2 plants-12-01505-t002:** The application of artificial intelligence and gene expression programming in tissue culture.

Modeling	Plant	Optimal Results
Multilayer perceptron (MLP) as an artificial ANN and support vector regression (SVR)	*chrysanthemum*	The highest embryogenesis rate (99.09%) and the maximum number of somatic embryos per explant (56.24) can be obtained [3]
Artificial neural network–genetic algorithm	*Garnem*	The results showed that the optimized rooting medium was more effective than the other standard medium [74]
GEP and M5′ model tree	*Pear* rootstocks	Proliferation rate, shoot length, shoot tip necrosis, vitrification and quality index; GEP had a higher prediction accuracy than the M5′ model tree [74]
Adaptive neuro-fuzzy inference system–genetic algorithm	*Corylus avellana*	Cell culture-responsive taxol biosynthesis was modeled and predicted by cell extract, culture filtrate, and cell wall alone or in combination with methyl-β-cyclodextrin [86]
Image processing and ANN	*Lycopersicon esculentum* L.	Plant growth regulators, the concentration of gum Arabic (GA) additive, the cold pretreatment duration, and flower length on callus induction percentage and number of regenerated callus in an anther culture of tomato [78]
ANNs-GA	*Pistacia vera*	Gain insights, predict, and optimize the effect of several independent factors on four growth parameters [79]

**Table 3 plants-12-01505-t003:** The effects of different types of nano-elicitors on plant tissues.

Classification of Nanomaterials	Size/Concentration	Source of Plant Materials	Results
AgNPs	0, 25, 50, 100 nm, and 200 ppm	*Rosmarinus officinalis* L.	Increased carnosic acid (CA) levels by more than 11% [104].
FeNPs	75 mg/L	Hairy-root of *Dracocephalum kotschyi*	Rosmarinic acid (RA) increased by 9.7-fold, xanthomicrol, cirsimaritin, and isokaempferide increased by 11.87, 3.85, and 2.27-fold, respectively [105].
Chitosan encapsulated zinc oxide nanocomposite	10–50 nm,	*Capsicum annuum*	Photosynthetic pigments (about 50%), proline (about 2 times), proteins (about 2 times), antioxidant enzyme activity (about 2 times), PAL activity (about 2 times), soluble phenols (40%), and alkaloids (60%) [106].
Single-wall carbon nano tubes	25, 50, 100, 125, and 250 μg/mL	Callus of *Thymus daenensis*	Total phenolic (TPC) content increased by 1.290 ± 0.19 mgGAEg^−1^ DW, total flavonoid (TFC) content increased by 2.113 ± 0.05 mgGAEg^−1^ DWimproved [107].
Graphene-based nanomaterials (GBNs)	50, 100, and 150 mg/L	* Ganoderma lucidum *	All GBNs increased the ganoderic acid (GA) content [108].
Magnetite nanoparticles (MNPs)	10.77, 20.5, 29.3 nm and 0.5, 1, 2 ppm	Callus of *Ginkgo biloba* L.	2 ppm + 10 nm MNPs increased the content of quercetin, kaempferol, p-coumarin, luding, caffeic acid, ginkgolide A, etc. [109].
SiO_2_NPs	2 mM	*Crocus sativus* L.	Increased the content of crocin and activity of superoxide dismutase (SOD), catalase (CAT), and ascorbate peroxidase (APX) [110].
Nano-TiO_2_	10, 60, and 120 mg/L	Callus of *Salvia tebesana*	The combination with methyl jasmonate increased the total phenols. O-diphenols, phenolic acid, flavonoid, flavane, flavonol, and proanthocyanidin were all increased [111].
TiO_2_/perlite nanocomposites (NCs)	15.50–24.61 nm	Callus of *Hypericum perforatum*	Induced the production of hypericin, pseudohypericin, and volatile compounds [112].
MgO/perlite nanocomposites (NCs)	10–30 nm	*Melissa officinalis*	Elevated volatile compounds. The new compound rosmarinic acid was detected [101].

**Table 4 plants-12-01505-t004:** Different chitosan nanomaterials as transport carriers in plant tissue culture.

Chitosan Nanomaterials and Carrier Materials	Dimensions (MD)	Results
Novel chitosan/alginate microcapsules simultaneously loaded with copper(II)cations and trichoderma viride	-	Chitosan/alginate microcapsules may incorporate both viral spores and chemical bioactivators without inhibiting their activity [134].
Deoxycholic acid carboxymethyl chitosan (DACMC) loaded with rotenone	91.3–140.0 nm	The in vitro release data of rotenone-loaded DACMC followed the Ritger and Peppas Case II transport mechanism. Highlights the potential of DACMC to reduce the use of organic solvents in the production of water-insoluble pesticides [136].
Alginate/chitosan nanoparticles encapsulated GA3	472–503 nm	Nanoparticles can improve the biological activity of gibberellic acid and have good application prospects in agriculture. Good performance and time stability [137].
Nanocarriers of plant growth regulator gibberellic acid (GA3) composed of alginate/chitosan (ALG/CS) and chitosan/tripolyphosphate (CS/TPP)	450 ± 10 nm	ALG/CS-GA3 nanoparticles have higher stability and efficiency in increasing the leaf area and chlorophyll and carotenoid content [97].
SA-CS NPs	368.7 ± 0.05 nm	SA-CS NPs can significantly affect the source activity by slowly releasing SA to manipulate various physiological and biochemical reactions of wheat plants [131].
SA-CS NPs	-	The results showed that the activity of antioxidant defense enzymes in maize increased, and the balance of reactive oxygen species (ROS) and the deposition of cell wall lignin increased, which had a positive effect on disease control and *maize* plant growth. It is a potential biological promoter [138].
Silica or chitosan encapsulated salicylic acid (SA) capsules	9.6–11.0 mm, 7.2–8.5 mm	In the in vitro system, the plants treated with a low proportion capsule had the best antifungal effect. At the same time, the capsule treated plants had higher levels of root and rosette development than the free SA treated plants [139].
Chitosan nanoparticles loaded with indole-3-acetic acid (IAA)	149–183 nm	CNPs-IAA can be applied to the hydroponic crop of crocantela variety *Latuca sativa* L. and has beneficial effects on plant growth, increasing the number of lettuce leaves by 30.9% [140].

## Data Availability

No new data were created or analyzed in this study. Data sharing is not applicable to this article.

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
