# Peer review of "Application of Data Modeling, Instrument Engineering and Nanomaterials in Selected Medid the Scientific Recinal Plant Tissue Culture"

_plants, 2023, doi:10.3390/plants12071505_

Round 1
Reviewer 1 Report
Dear Authors,
congratulations on a review that covers a very interesting topic comprehensively and with consistent reinforcement based on the existing bibliography.
I'm giving only some suggestions for improvement of some sections of the manuscript:
1) in section 2 some comment could be included on the risks of somaclonal mutations when high doses of auxin substances are applied for callus induction. Indeed, it is widely known that this aspect has often limited the application of tissue culture models through callus induction and subsequent embryogenesis.
2) In the same section, moreover, when discussing the effects of tissue culture in the emission of primary and secondary roots, the determinant effects of light are only mentioned while they certainly deserve a little further investigation since much work has been done on these aspects in the international scientific literature.
3) In section 2.2.1., a table could be included in which all PGRs are listed with a clear indication of their roles, functions ascribed by science, and reference literature. This would clearly define the knowledge and differences between old and new PGRs.
4) In defining the role of tissue culture and the use of bioreactors one should also report the case of the presence of natural endophytes that can greatly influence the efficiency of temporary immersion with potential (usually bacterial) pollutions to be controlled differently. This aspect often conditions the possibility of developing tissue accretion models through the use of bioreactors.
5) It would probably be useful to include a graphical model that can define the role of artificial intelligence as reported in the scientific literature as well as that of gene expression programming, both of which are valuable approaches in the study and validation of models of tissue culture use for secondary metabolite production.
Your review is very well articulated and my suggestions would, in my opinion, serve to add to its completeness.
Author Response
Please see the word

Reviewer 2 Report
Dear Authors,
The article is interesting and presents, among other things, innovative techniques used in plant tissue cultures. Unfortunately, the manuscript is slightly chaotic, with many editorial and substantive errors so I think that the Authors should take count a modification of this article. I recommend publishing it in "Plants" after a major revision.
Due to the fact that the manuscript submitted for review does not include line numbering, I included detailed comments directly in the file.
I would like to make two general remarks here:
1. in my opinion, the "Discussion" chapter in this type of article has no reason to be, it is better to replace it with a chapter, e.g. "Conclusion" or "Future perspective"
2. Authors should review the manuscript very carefully in an editorial context
With best regards

Author Response
Response to the Reviewers
Thank you very much for your kind review and give us good revision comments. We have carefully revised the manuscript according to the reviewers’ suggestions. The revised portions have been labeled in red font in the resubmission manuscript.
Reviewer(s)' Comments to Author:
Reviewer 2:
Comments to the Author
Question 1: in my opinion, the "Discussion" chapter in this type of article has no reason to be, it is better to replace it with a chapter, e.g. "Conclusion" or "Future perspective"
Thank you for your advice ,We have modified, if there is a problem, we can modify again.
Question 2: Authors should review the manuscript very carefully in an editorial context
Thank you very much for your careful comments and apologize for our carelessness. We have modified the annotation part in the corresponding position in the text, such as the abbreviation form, the table format error, the language is not rigorous and so on.
If there are still inappropriate places, welcome to point out.we would like very much to modify them and we really appreciate your help.

Round 2
Reviewer 2 Report
Dear Author,
In my opinion, the Authors have significantly improved the manuscript and after a slight correction (beneath) in the corrected form, it can be accepted for publication in Plants.
1. I suggest also reading the manuscript text carefully again to eliminate minor typing errors and standardize subsection headings, standardize text indentation, and refine figure captions and literature cited in the tables.
2. Species and genera names should be written in italics
Comments:
Abstract:
Line 26 – 27: “It can help to improve the tissue culture technology of medicinal plants and increase the yield of natural metabolites” – I suggest: “It can help to improve the tissue culture technology of medicinal plants and increase the availability and the yield of natural metabolites”
Introduction:
Line 108: “The cell in callus also has totipotency ……” – language awkwardness, I suggest: “In callus mass, there are some totipotent cells”
Line 108 - 109: “In vitro culture, callus is mostly induced by auxin and cytokinin. – I suggest: “ In in vitro culture or: In vitro callus is induced by the synergistic action of auxins and cytokinins”
Line 109 – 111: “High proportion of 1H-Indole-3-acetic acid (IAA) is good for root regeneration[29], and a high proportion of cytokinin is good for stem regeneration” – I suggest something like this: “The mutual correlation between auxins and cytokinins affects regenerative processes in in vitro cultures. The high concentration of Indole-3-acetic acid (IAA) promotes root formation while high proportion of cytokinin is good for adventive shoots regeneration [30]”
Line 119: “Trauma ……….” – I suggest to add: “Mechanical trauma or mechanical damage ….”
Line 119: factors[å¼ 2][å¼ 3] - cancel
Line 137: “.… in vitro.” – I suggest : “…….. in in vitro conditions”
Line 147 – 148: “The genes that regulate callus and lateral roots are called 147 Solitary Root (SLR) – it is necessary to explain what parameters they regulate and in the same line Authors have to decide “dott” or “coma”?
Line 160 – 162: if this text plays the function of figure 2 description it should be it should be unambiguously linked to an existing signature
Line 169: cytokinis - spelling
Line 566: [å¼ 4] - ?
With best regards,
Author Response
Thank you very much for your kind review and give us good revision comments. We have carefully revised the manuscript according to the reviewers’ suggestions. The revised portions can be find in the resubmission manuscript.
Reviewer(s)' Comments to Author:
Reviewer 2:
Comments to the Author
Question 1: I suggest also reading the manuscript text carefully again to eliminate minor typing errors and standardize subsection headings, standardize text indentation, and refine figure captions and literature cited in the tables.
Thank you very much for your careful comments .We have modified the annotation part in the corresponding position in the text, please see the file uploaded.
Question 2: Species and genera names should be written in italics
Thank you for your advice,the above problems have been corrected in the text.
If there are still inappropriate places, welcome to point out.we would like very much to modify them and we really appreciate your help.
